# Subdural Effusion Evolves into Chronic Subdural Hematoma after Deep Brain Stimulation Surgery: Case Report and Review of the Literature

**DOI:** 10.3390/brainsci12101375

**Published:** 2022-10-10

**Authors:** Dongdong Wu, Yuanyuan Dang, Jian Wang, Zhiqiang Cui

**Affiliations:** Department of Neurosurgery, The First Medical Clinical Center, PLA General Hospital, Beijing 100853, China

**Keywords:** chronic subdural hematoma (CSDH), subdural effusion (SDE), deep brain stimulation (DBS)

## Abstract

Background: Although chronic subdural hematoma (CSDH) has been known for over several hundred years, the etiology and pathogenesis of it are still not completely understood. Neurosurgical procedures resulting in CSDH are a rare clinical complication, and there was no report about how subdural effusion (SDE) evolves into CSDH after deep brain stimulation (DBS) surgery. The formation mechanism of CSDH after surgery, especially in DBS surgery, and the effect of recovery, need to be explored. Methods: We present two cases, complicated with SDE after DBS surgery, serious dysfunction complications such as hemiplegia and aphasia occurred on the postoperative day 36 and 49 individually, and images showed CSDH. Fusion image showed the bilateral electrodes were significantly shifted. Then, they were performed to drill craniotomy with a closed system drainage. Result: The symptoms of hemiplegia and aphasia caused by CSDH were completely recovered, and the follow-up images showed CSDH was disappeared. However, DBS stimulation is poorly effective, it cannot reach the preoperative level, especially in the ipsilateral side of CSDH. Conclusions: The iatrogenic SDE that evolved into CSDH in the present two cases shows that SDE is one of the causes of CSDH. Patients develop SDE after DBS, which increases the risk of developing CSDH. CSDH after DBS can be successfully treated. however, the postoperative efficacy of DBS will decline.

## 1. Introduction

The first case of chronic subdural hematoma (CSDH) was reported in 1657 [1], although the term CSDH was not introduced until 1925 [2]. Despite CSDH being reported for several hundred years, the etiology and pathogenesis of CSDH are still not completely understood [3,4,5,6,7,8]. The etiologies of CSDH include chronic alcoholism, long-term anticoagulation (e.g., aspirin, heparin, or warfarin), spontaneous leakage of cerebrospinal fluid (CSF), and neurosurgical procedures (e.g., ventricular shunting or craniotomy), which can lead to intracranial pressure fluctuations and stretching of the delicate vasculature [9,10,11,12,13,14,15]. There are several hypotheses about the pathogenesis of CSDH, including the neoformation of a subdural membrane, abnormal vascular permeability, defective local hemostasis (hyperfibrinolysis), tearing of a bridging vein by minor trauma with subsequent chronic rebleeding, and/or traumatic subdural hygroma (SDG) [3,8,16,17,18,19].

So far, there has rarely been reports on CSDH after DBS surgery. Herein, we present two cases in which CSDH occurred after deep brain stimulation (DBS), with evidence that subdural effusion (SDE) evolved into CSDH. The iatrogenic SDE evolving into CSDH shows that SDE is one of the causes of CSDH, and also clarifies the partial mechanisms of CSDH formation. We also analyzed the cause of CSDH after DBS surgery and evaluated its impact on the patients.

## 2. Case Reports

### 2.1. Case 1

Patient 1 was a 62-year-old right-hand-dominant male with a history of a resting tremor in both extremities since the age of 52 years. He had been diagnosed with Parkinson’s disease, for which he had been taking oral medication. However, the symptoms of tremor and stiffness gradually worsen, and develop to both limbs, and the drug effect gradually decreases after 5 years. The patient developed peak-dose dyskinesia and the ‘wearing off’ phenomenon after 8 years. The patient had a Unified Parkinson’s Disease Rating Scale (UPDRS III) motor score of 69, Mini-Mental State Examination (MMSE) score of 27, Montreal Cognitive Assessment score of 24, 39-item Parkinson’s Disease Questionnaire score of 38, and Hoehn and Yahr score of 4. The L-dopa stimulation test showed a 55% improvement. Conventional brain MRI showed mild atrophy and dilation of the ventricles (Figure 1a). The patient then underwent posteroventral globus pallidus internus (GPI) DBS at the age of 62 years. The surgical procedure is described in our previous study [20,21].

Under local anesthesia, a stereotactic head frame (Leksell model F head frame) was placed before stereotactic CT was performed. CT data was fused with the preoperative plan that was developed using MRI performed with a 3.0-T scanner (Siemens Espree). The anatomical target coordinates for STN stimulation were 12.5 mm lateral, 2.5 mm posterior, and 5 mm inferior to the midpoint of the anterior commissure-posterior commissure (AC-PC) line. To reduce CSF leakage, the neck was flexed to raise the head as far as possible while maintaining airway patency. The STN coordinates were routinely refined using intraoperative microelectrode recording (MER). Three-track MER was obtained using the Alpha-Omega neural activity monitoring system (Alpha Omega Engineering Ltd., Nazareth, Israel). The microelectrode was advanced to 8 mm above the target, and one recording was made on each side to obtain the typical STN signal. During the recordings, the burr holes were covered with fibrin sealant to reduce the loss of CSF. However, some CSF was still lost because of the bilateral widened subarachnoid spaces. After MER and placement of the DBS PINSL301 electrodes (PINS Medical Co., Beijing, China), test stimulation was conducted using a temporary external stimulator. The patient remained awake so that the temporary efficacy could be observed, and verbal feedback could be obtained to ensure that unwanted adverse effects did not occur.

The intraoperative MRI showed no subdural effusion in bilateral frontal areas and the accuracy of the electrode positioning (Figure 1b). On postoperative day 5, CT showed a small amount of SDE and pneumocephalus in the right frontal lobe (Figure 1c); the patient had no changes in neurological symptoms and signs before starting the stimulation. On postoperative day 20, brain pacemaker therapy was commenced. Stimulation of the GPI resulted in a UPDRSIII motor score of 30, which was a 56% improvement compared with preoperatively, and there were no adverse effects.

At 36 days postoperatively, the patient had slow speech and mild left hemiparesis, and the efficacy of DBS was markedly reduced compared with the initial stimulation. Although we increased the amplitude, the UPDRS motor score was 81. The patient denied ever experiencing head injury or alcohol addiction. CT and MRI showed a CSDH, and the bilateral electrodes were clearly shifted to the left (Figure 1c,d). We immediately stopped the electrical stimulation. A drilling craniotomy was performed in the right parietal skull with closed system drainage. After 2 days of drainage, CT revealed that most of the hematoma on the right side had disappeared, the midline was centered, and the electrodes had returned to their original positions (Figure 1e). We measured the volume of CSDH with 3D slicer software, which provides a free open-source software platform for biomedical research to be conducted (http://www.slicer.org). The change in CSDH volume in pre- and post-operative evacuation can be seen in Figure 2. Comparison with the early CT images showed that the electrode was no longer shifted. The language dysfunction and limb paralysis recovered within 1 week postoperatively. Deep brain stimulation was resumed 4 weeks after drainage. The UPDRS III score was 58.

### 2.2. Case 2

Patient 2 was a 56-year-old right-hand-dominant male who presented at our outpatient clinic due to a gradual cognitive decline that had started 3 years earlier. There had been severe memory decline over the past 13 months; the patient frequently lost daily supplies, could not remember his phone number, and was sometimes unable to find his way home. Treatment with donepezil HCl had been ineffective. The patient had a MMSE score of 16, and an Alzheimer’s Disease Assessment Scale-cognitive subscale (ADAS-Cog) score of 38. Moreover, 18-Florbetaben PET images revealed bilateral temporal-parietal lobe hypometabolism. CSF analysis was normal. Brain MRI demonstrated global cerebral atrophy (Figure 3a,b). Assessment by three neurologists resulted in a diagnosis of Alzheimer’s disease. Bilateral DBS of the fornix and hypothalamus was performed. The surgical procedure is described in our previous study [20,21].

The surgical procedure was the same as that of the first patient, with the following differences. Surgery was performed under general anesthesia. The anatomical target coordinates for bilateral fornix-hypothalamus stimulation were 5 mm lateral, 8 mm anterior, and 7 mm inferior to the midpoint of the AC-PC line. The final trajectories and lead placements were accepted after autonomic effects (including blood pressure elevation and a sensation of warmth) could be demonstrated, beginning at approximately 5–7 V of stimulation. Single-track MER (Medtronic Ltd. Minneapolis, MN, USA) was performed with DBS PINSL302 electrodes (PINS Medical Co. Beijing, China).

On postoperative day 1, CT showed bilateral frontal pneumocephalus and mild bilateral SDE, more obvious on the left side (Figure 4a). MRI on postoperative day 5 showed right frontal SDE (Figure 3c–f). A comparison of the CT data with the preoperative plan showed that the electrodes were accurately positioned in the bilateral fornix-hypothalamus. At 14 days postoperatively, brain pacemaker therapy was commenced. The symptoms of memory loss were relieved for 20 days. The patient and his wife were satisfied with the outcome. The MMSE score was 20, and the ADAS-Cog score was 30.

On postoperative day 49, the patient’s speech function worsened, and he developed mild right hemiplegia and a sudden worsening of cognitive function. The patient and his wife denied that a head injury had occurred. CT showed a CSDH, and a right shift of the midline and the bilateral electrodes (Figure 4b). Fused with the previous CT images, the maximum shift distances in the paths for the right and left electrodes were 2.7 mm and 5.7 mm, respectively (Figure 4c,d).

We immediately stopped the electrical stimulation. The patient underwent a drilling craniotomy in the left parietal skull with closed system drainage. The limb paralysis recovered within 3 days postoperatively, while the language dysfunction recovered more slowly. CT revealed that most of the hematoma on the left side had disappeared after 3 days of drainage, the midline had centered, and the left electrode had returned to its original position. Fused with the early CT images showed that the electrode was no longer shifted (Figure 4e). At 77 days postoperatively, CT showed complete disappearance of the hematoma (Figure 4f). The change in CSDH volume in pre- and post-operative evacuation in case 2 see Figure 5. Deep brain stimulation was resumed 4 weeks after drainage. However, after 2 months of drainage, the MMSE and ADAS-Cog scores had not returned to the initial values.

## 3. Systematic Review

A systematic review was performed, applying the PRISMA (Preferred Reporting Items for Systematic Reviews and Meta-Analyses) guidelines [22]. Full-text articles were selected from a comprehensive search of PubMed, Medline, Scopus and Google Scholar databases. Keywords and their synonyms were combined in each database as follows: (“deep brain stimulation”) AND (“chronic subdural hematoma”). No filter was applied on the publication date of the articles, and all results of each database were included up to January 2022. After the removal of duplicates, all articles were evaluated through a screening of titles and abstracts by three independent reviewers (Y.D., J.W., Z.C.). These researchers read the full-text articles deemed suitable for the study and performed data collection to reduce the risk of bias. In case of disagreement among the investigators regarding the inclusion and exclusion criteria, the senior investigator (Z.C.) made the final decision.

The inclusion criteria were: (i) Chronic subdural hematoma without inducing factors within 6 months after DBS; (ii) English language; and (iii) published in a peer-reviewed journal. The exclusion criteria were: (i) Acute subdural hematoma after DBS operation; (ii) Chronic subdural hematoma caused by trauma more than 6 months after DBS operation; and (iii) conference proceedings, or reviews and books.

### Data Extraction Process

Data extraction was executed on 8 articles (Figure 6).

A total of 2 articles were excluded because of duplicates. Moreover, three articles were excluded, this is because two of them were acute subdural hematoma during DBS operation, and the other one was acute subdural hematoma after head trauma in DBS postoperative 2 years. After an accurate revision of full manuscripts, 3 articles satisfied the inclusion/exclusion criteria (Figure 6).

## 4. Results

Through the literature search and screening, only 3 articles reported the CSDH after DBS, however, the description of the disease is not detailed, lacking the gender, age, disease duration and the diagnosis of the disease. The target of DBS was also unclear, and the image data before operation was not detailed. The average time of the evacuation in the three patients is 2 months, and the impact of CSDH on the position of the lead and the long-term stimulation effect are also unclear [23,24,25]. Two authors analyzed the possible causes of CSDH, including brain atrophy, excessive cerebrospinal fluid loss, pneumocephalus etc. [24,25] (see Table 1). Since the number of cases is very small, and the data is not comprehensive, statistical methods cannot be used to analyze the relevant factors of the CSDH.

## 5. Discussion

Intracerebral hemorrhage is the most severe complication of DBS surgery. The incidence of intracerebral hemorrhage in DBS surgery is reportedly 0.2–5.6% [26,27,28,29,30]. Neurosurgical procedures reportedly only result in CSDH in 0.8% of patients [31], while the occurrence of CSDH after DBS surgery has only been reported in three cases. The reasons for CSDH after DBS have not been reported [23,24,25].

One of the potential complications after DBS surgery is SDE. Most SDE associated with brain surgery are self-absorbed, with only few reports of the conversion of iatrogenic SDE into CSDH [15]. There are no previous reports of SDE evolving into CSDH after DBS surgery.

The cause of SDE after DBS is unknown. Our two cases had the following similarities: (1) preoperative cerebral atrophy (especially in patient 2); (2) large loss of CSF, leading to low intracranial pressure; (3) frontal pneumocranium several days after surgery; (4) SDE. Furthermore, the electrodes passed through the lateral ventricles in patient 2. This suggests that the potential causes of SDE in DBS include cerebral atrophy with widened subarachnoid spaces, opening of the lateral ventricles, CSF loss, and postoperative intracranial hypotension.

The present patients had no history of head trauma. However, the conventional explanation for CSDH is the tearing of a bridging vein, which is usually caused by minor trauma. In both patients, MRI revealed bilateral SDE and CSDH. The location of the CSDH was consistent with that of the SDE in both cases, suggesting that the SDE evolved into the CSDH. This is consistent with several reports that traumatic SDE is a predisposing cause of CSDH [3,8,16,17,18,19]. The main pathological mechanism by which SDE evolves into CSDH may be local inflammatory reactions, including the proliferation of immature capillaries, cytokines, local hyperfibrinolytic activities, and resultant bleeding, which ultimately increase the hemoglobin in the SDE [32,33,34,35].

Feng et al. [3] hypothesized that SDE and CSDH are two stages of the same inflammatory process. The concentrations of interleukin-6 and interleukin-8 were both elevated in SDE and CSDH compared with serum levels [3]; furthermore, the levels were much higher in CSDH than in SDE [3]. SDE and CSDH are fundamentally distinct regarding the appearance of their contents and the respective CT values, but not regarding pathogenesis. Our two cases suggest that the SDE evolved into CSDH, supporting the theory proposed by Feng et al. [3].

A previous study reported the diagnosis of SDE 11.6 days after injury. The average interval between the development of SDE and the evolution of CSDH is 101.5 days [36], or 22–100 days after a head injury [37]. The SDE developed on the first day after surgery in both of the present patients, and the respective times taken for the SDE to evolve into CSDH were 36 and 49 days. The relatively short time between SDE development and CSDH appearance may be related to the faster formation of postoperative SDE compared with traumatic SDE. The relatively fast formation of postoperative SDE was probably related to the cerebral atrophy in patient 1, the electrodes passing through the lateral ventricle in patient 2, and the large amount of CSF loss in a short period plus the low intracranial pressure for several days postoperatively in both patients.

The conversion of iatrogenic SDE into CSDH is not exactly the same as the conversion of traumatic SDE into CSDH (Table 2). Trauma cases have no imaging information from before the trauma, and so we cannot judge whether there was preexisting SDE and severe cerebral atrophy. The relationship between the change in SDE volume and the formation of CSDH has not been reported. The timing of SDE formation after trauma and the time taken for the SDE to change into CSDH are still unclear, as the number of reported cases is still very small. In the present cases, DBS surgery artificially caused a SDE that evolved into CSDH. The pre- and post-operative imaging data are complete and can partially explain the process of SDE transformation into CSDH. The limitations of these two cases include the lack of inflammatory cytokine testing, and the lack of interval imaging data between the development of SDE and CSDH.

Hasegawa et al. [38] reported meningeal enhancement on MRI with Gd-DTPA enhancement in five patients with traumatic SDE. Microscopic examination of the enhanced dura mater revealed a vascularized neomembrane with numerous pinocytic vesicles and fenestrations in the vessel endothelium, suggesting that SDE with meningeal enhancement has the potential to develop into CSDH [38]. Similarly, patients with severe cerebral atrophy may develop SDE after DBS surgery, which is a risk factor for CSHD formation.

The formation of a moderate or larger amount of CSDH can cause obvious symptom changes and a reduction in the efficacy of DBS stimulation. After CSDH formation, the limb stiffness and tremor in patient 1 returned to preoperative levels, while patient 2 showed reduced speech and slow responses. However, such changes are often not severe enough to alert the patient’s family and the doctor to the possibility of CSDH; hence, the initial decrease in the effectiveness of the DBS caused the doctors to adjust the stimulation parameters by increasing the voltage in both of the present cases. It was not until the development of very obvious symptoms (limb hemiplegia and speech disorder) that CT was performed, and CSDH was diagnosed in both patients.

The large amount of hematoma in the present two cases caused the electrodes to shift so that the contacts were stimulating non-target brain tissue; this results in wasted current, as the stimulus is invalid, and so we stopped the stimulation immediately. More importantly, serious dysfunction complications such as hemiplegia and aphasia occurred, necessitating emergency hematoma evacuation.

CSDH is generally effectively treated via burr hole craniotomy with closed system drainage. The hematoma will then either completely disappear, or a small amount of residual hematoma fluid can be slowly absorbed by the administration of oral atorvastatin [39]. For patient 1, the hematoma was completely absorbed, and there was complete resolution of the limb hemiplegia and speech dysfunction. However, although the electrode on the contralateral side of the hematoma had no obvious shift and subsequent stimulation achieved the original effect, the effect of stimulation on the ipsilateral side did not reach the preoperative level, even after the hematoma was completely absorbed and the fusion images showed no shift in electrode position.

The most important factor in minimizing the risk of CSDH after DBS is to reduce the formation of SDE. Hence, DBS treatment should be performed cautiously in older adult patients with obvious cerebral atrophy and widened subarachnoid spaces. Furthermore, the intraprocedural CSF loss should be minimized; to maintain intracranial pressure stability under general anesthesia, the head should be maintained in a high position, the operation time should be as short as possible (by reducing the MER time and the number of punctures), bio-adhesive sealant should be applied to cover the dural holes, and the puncture path should avoid the ventricles. Regarding the pneumocephalus and CSF loss in DBS surgery, Krauss et al. has introduced their experience, which we can learn from [40]. Older adult patients with cerebral atrophy should have a low head position, and extended bedrest postoperatively. Routine head CT or MRI examinations can enable the early detection of SDE. Finally, if the postoperative effect of stimulation gradually decreases even when the stimulation parameters are increased, clinicians should consider the possibility of SDE and CSDH, and imaging information should be obtained, as early detection and treatment is beneficial for recovery.

It is extremely rare for CSDH to develop after DBS surgery, with only three cases reported previously [23,24,25]. Panov et al. [23] reported that CSDH formed on the side of the cortical electrode in 1 of 200 (0.5%) patients, and that burr hole evacuation at 2 months after DBS implantation resulted in no neurological sequelae; however, they did not report whether there was postoperative SDE, a shift in the electrode position, and/or the long-term stimulation effect. Ponce et al. [24] also reported one patient with Alzheimer’s disease who presented 60 days postoperatively after a ground-level fall that caused a clavicle fracture and was found to have bilateral CSDH with a mass effect. Along the parasagittal frontal lobe, which was the entry point for the DBS lead, there was about 1 cm of CSF between the skull and the gyrus, and the patient had marked pneumocephalus postoperatively [24]. The hematomas were believed to be a consequence of the pneumocephalus, and hematoma evacuation resulted in complete resolution of the hematoma [24]. The description of this previous case suggests that the CSF between the skull and the gyrus was actually SDE, which had evolved into CSDH by 60 days postoperatively. We consider that the CSDH was probably caused by SDE, rather than pneumocephalus. It was not reported whether there was a shift in the electrode position, and/or a change in the long-term stimulation effect. Umemura et al. [25] reported that one patient with Parkinson’s disease developed postoperative CSDH, most likely as a result of cerebral atrophy and excessive outflow of CSF during DBS; the patient showed severe hemiparesis and required evacuation of the hematoma 2 months after DBS implantation [25]. The position of the DBS lead had not moved, and replacement was not indicated [33]. These are the only five patients in whom CSDH after DBS surgery was reported (Table 1). Although these previous cases contain incomplete data, the common characteristics include cerebral atrophy, loss of CSF, SDE, and low intracranial pressure. However, we believe that SDE is not the only cause of CSDH after DBS.

## 6. Conclusions

The iatrogenic SDE that evolved into CSDH in the present two cases shows that SDE is one of the causes of CSDH. Patients with severe cerebral atrophy develop SDE after DBS, which increases the risk of developing CSDH. Large CSDH formed after DBS can be successfully treated. The DBS electrode positions undergo a small shift after drilling craniotomy with closed system drainage; however, DBS stimulation is poorly effective, as it is difficult to reach the preoperative level.

## Figures and Tables

**Figure 1 brainsci-12-01375-f001:**
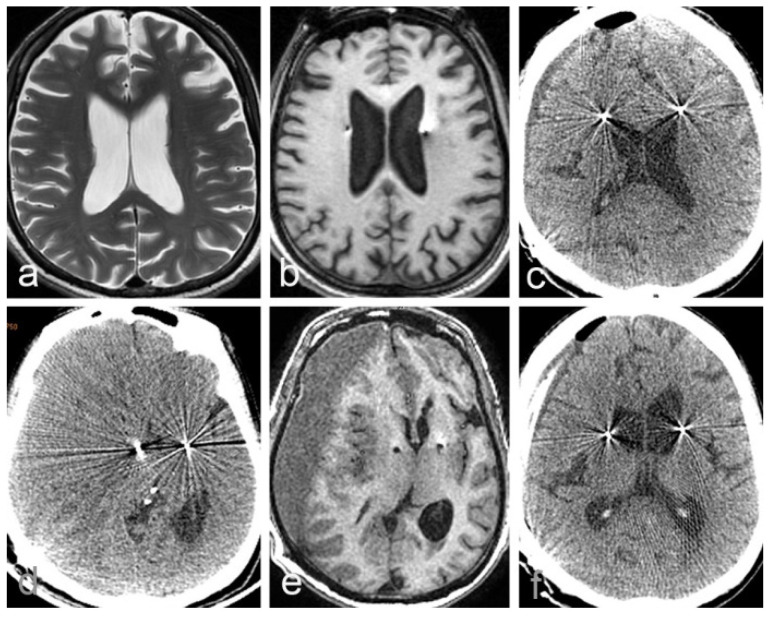
The change in MRI and CT images of the patient 1. (**a**) Conventional brain MRI, T2 image showed mild atrophy and dilation of the ventricles; (**b**) The image of the intraoperative 3 dimensional T1-weighted sequence showed no subdural effusion in bilateral frontal areas; (**c**) At 5 day after DBS surgery, CT shows a small amount of subdural effusion and pneumocephalus in the right frontal lobe; (**d**,**e**) At 36 days after DBS surgery, CT and T1 MRI images of the brain shows right chronic subdural hematoma, with the midline obviously shifted to the left, and a marked shift in electrode positioning; (**f**) At 2 days after drainage, CT shows that most of the hematomas were drained out and the electrode shift has been corrected.

**Figure 2 brainsci-12-01375-f002:**
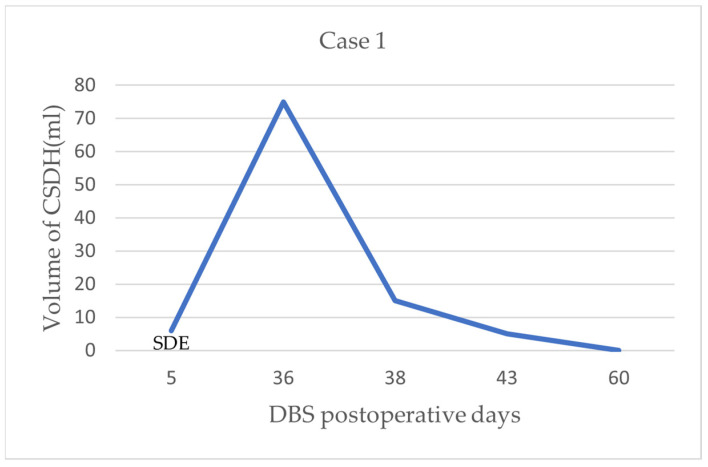
The change in CSDH volume in pre- and post-operative evacuation in case 1. At 5 day after DBS surgery, CT shows the volume of subdural effusion (SDE) in the right frontal lobe was 6 ml; at 36 days after DBS surgery, the volume of right chronic subdural hematoma is 75 mL; at 2 days after drainage, residual hematomas volume was 15 mL; until day 60, the hematoma was completely absorbed.

**Figure 3 brainsci-12-01375-f003:**
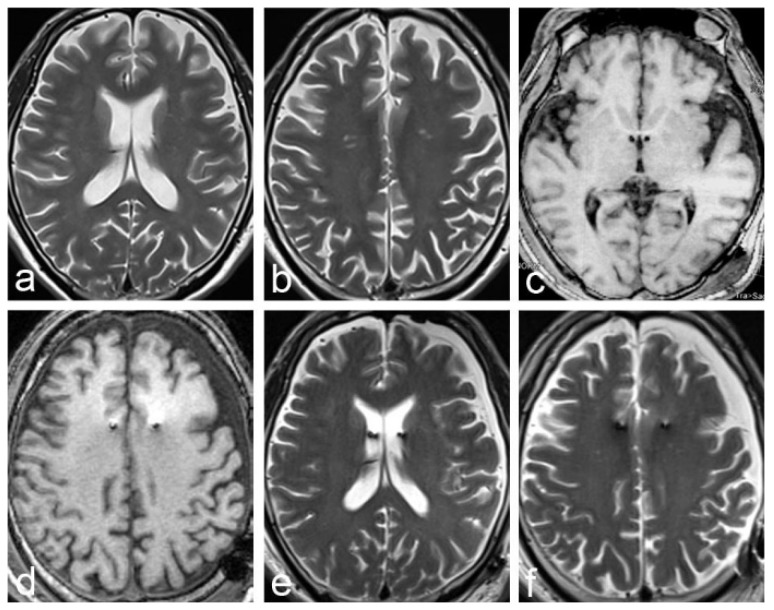
MRI images in patient 2 before the formation of chronic subdural hematoma. (**a**,**b**) Preoperative MRI shows cerebral atrophy, and that the left subarachnoid space is wider than the right; (**c**–**f**) Five days postoperatively, MRI (3DT1 and T2 image)shows bilateral subdural effusion that is more obvious on the left side. The lateral ventricle is slightly reduced, and the fornix-hypothalamus electrode passes through the bilateral lateral ventricles.

**Figure 4 brainsci-12-01375-f004:**
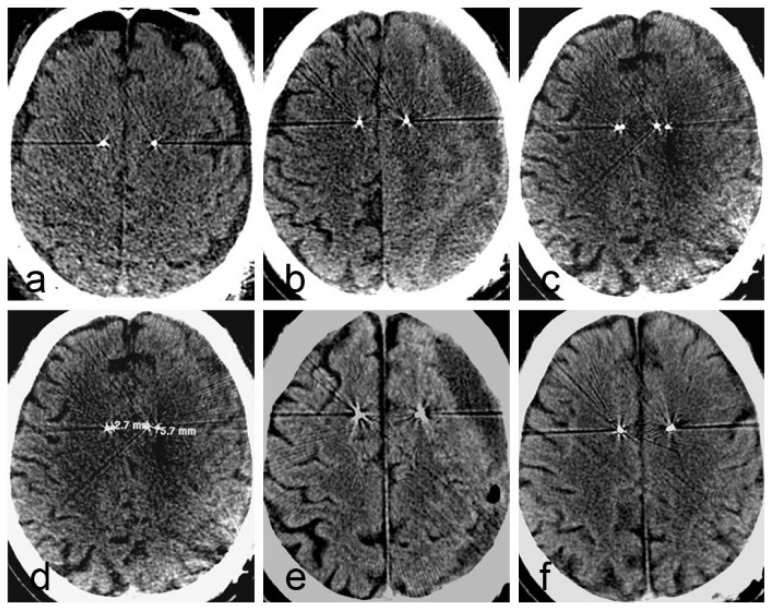
The change in CT images in patient 2. (**a**) On the first day after deep brain stimulation (DBS), CT shows bilateral subdural effusion and pneumocephalus, more obvious on the left side; (**b**) At 49 days after DBS, CT shows a CSDH on the left side, and a slight contralateral shift of the midline; (**c**,**d**) Fused with the early CT, the bilateral electrodes are substantially shifted; (**e**) At 3 days after drainage, CT fusion shows that the electrode shift is corrected; (**f**) At 77 days after drainage, CT shows complete hematoma absorption.

**Figure 5 brainsci-12-01375-f005:**
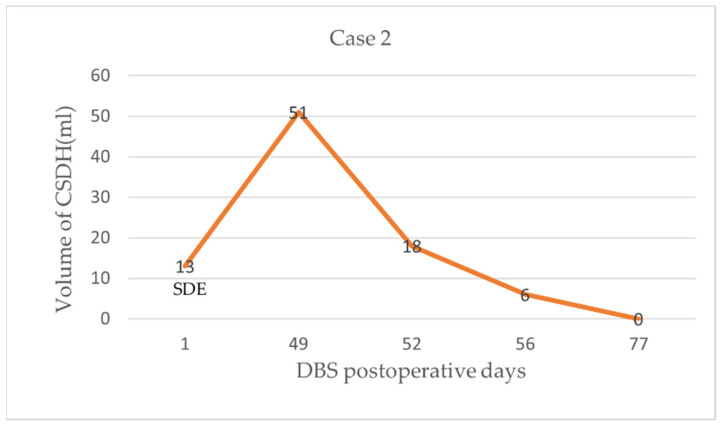
The change in CSDH volume in pre- and post-operative evacuation in case 2. At 1 day after DBS surgery, CT shows the volume of subdural effusion (SDE) in the left frontal lobe was 13 mL; at 49 days after DBS surgery, the volume of right chronic subdural hematoma is 51 mL; at 3 days after drainage, residual hematomas volume was 18 mL; until day77, the hematoma was completely absorbed.

**Figure 6 brainsci-12-01375-f006:**
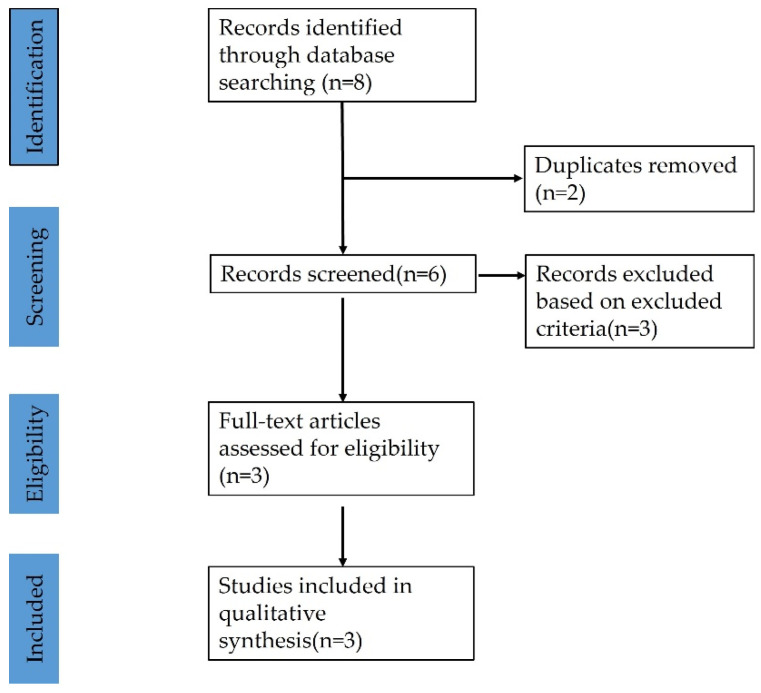
PRISM flowchart of the selection of the studies for this review.

**Table 1 brainsci-12-01375-t001:** The characteristics of the five reported patients with CSDH after DBS surgery.

No.	Age (Year)	Sex	Diagnosis	DBS Target	SDE	Cerebral Atrophy	Days of Evacuation	Lead Shift	Effect of Stimulation
Case 1	62	F	Parkison’s disease	GPI	Yes	Yes	36 days	description	poor
Case 2	56	M	Alzheimer’s disease	fornix	Yes	Yes	49 days	description	poor
Panov et al. 2017 [23]	—	—	Movement disorder	—	—	—	2 months	—	—
Ponce et al. 2016 [24]	—	—	Alzheimer’s disease	fornix	Yes	—	60 days	—	—
Umemura et al. 2011 [25]	—	—	Parkinson’s disease	STN	—	Yes	2 months	No shift	—

“—” indicates that there was no description; DBS: deep brain stimulation; GPI: globus pallidus internus; STN: subthalamic nucleus; SDE: subdural effusion.

**Table 2 brainsci-12-01375-t002:** Comparison of two types of subdural effusion (SDE).

Arachnoid	SDE after DBS	Traumatic SDE
Breakage	Clear	Unclear, Inference
Location of breakage	Bilateral forehead	Unclear location
Broken shape	Round hole	Irregular shape
Whether there is a valve	No	Possible

## Data Availability

Not applicable.

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
