# Peer review of "Subdural Effusion Evolves into Chronic Subdural Hematoma after Deep Brain Stimulation Surgery: Case Report and Review of the Literature"

_brainsci, 2022, doi:10.3390/brainsci12101375_

Round 1
Reviewer 1 Report
The authors present an interesting article on subdural effusions and consecutive hematomas after deep brain stimulation interventions. Both the topic and content of the manuscript are relevant for neurosurgeons as well as all other neuroscientists. Although rarely reported so far, in my opinion both subdural entities occur much more frequently after neurosurgical procedures, including deep brain stimulation. Nevertheless, there are some minor aspects that should be considered and worked out:
Case 1 presentation. I would recommend to briefly mention the medication as well as summarize the surgical procedure (even if the description in a previous article is referred to). Furthermore, I would not write "start machine" but from "start/initiate stimulation" etc. and would prefer "initial" to "first" stimulation. In addition, the term “deep brain stimulation” seems to be more appropriate than “brain pacemaker treatment” from a neuromodulatory point of view. What is meant by the phrase "we increased the stimulation parameters" (the amplitude)? There are no major abnormalities in the figures, but in my opinion at least mild atrophy and dilation of the ventricles.
Case 2 presentation. Again, the DBS procedure could be briefly described. It would be interesting to see whether the operation is purely image-based or whether a microelectrode recording with clinical stimulation (obviously not) was carried out in addition.
Discussion. Regarding the reference Hasegawa et al., the authors inadvertently write of SDG instead of SDE which should be corrected. The ratio behind atorvastatin treatment for chronic subdural hematoma and a literature reference could be added.
Table 2. The spelling of Parkinson needs to be corrected.
Conclusions. I would better replace the term "model" by a more appropriate term.
Regarding risk factors for postoperative intra-cranial air inclusions and subdural effusions, I recommend additionally the following reference: Krauss P, Van Niftrik CHB, Muscas G, Scheffler P, Oertel MF, Stieglitz LH.How to avoid pneumocephalus in deep brain stimulation surgery? Analysis of potential risk factors in a series of 100 consecutive patients. Acta Neurochir (Wien). 2021;163(1):177-84.
Author Response
请参阅附件。

Reviewer 2 Report
The authors reported rare complication of SDE and cSDH after DBS surgery. They performed also review of the literature, which gives the case report more impact. Thus, I would restructure the paper.
Titel: "Subdural effusion evolves into chronic subdural hematoma after deep brain stimulation surgery" - case report and review of the literatures
I would make a separate section: review of the literature. Then table 2 should be included in that section with explantation of material/methods and results. Table 2 should be revised. I would add the section authors and references should be added to it.
The authors performed several follow-up controls via CT or MRI. I would suggest to perform a volumetric analysis and add a figure with the change of volume (y-axis) according to postoperative days (x-axis).
Here are some minor issues:
Case1:
-"After 2 days of drainage, CT revealed that most of the hematoma on the right side had disappeared, the midline was centered, and the electrodes had returned to their original positions (Fig. 1e)" - I think that the postoperative CT scan is shown in Fig. 1f.
Case 2
-"MRI on postoperative day 5 showed right frontal SDE (Fig. 2c-f)." - I think the SDE is on the left side.
Reviewer 3 Report
I am not sure if these two cases are rare or not. A surgical procedure like craniotomy sometimes causes CSDE and CSDH. Clinically, CSDH is usually developed from CSDE. Burr hole surgery is less invasive than a craniotomy, which might explain the rare incidence of CSDH. Could you show the incidence rates of CSDH after craniotomy or other burr hole surgery?
I also cannot understand the importance of DBS surgery in explaining the authors' hypothesis. For example, over-drainage of CSF in shunt or drainage surgeries sometimes causes CSDE and, in the worst situation, CSDH. How many CSDH cases have been reported after coagulation surgeries like thalamotomy and pallidotomy? The authors should discuss more the relationship between other brain surgeries, especially burr hole surgeries, and CSDE/CSDH.
Finally, the important aspect of these cases was the decrease of the DBS effect even after the disappearance of CSDH. Did you check the exact position of the DBS leads? What were the long-term effects? If the DBS leads returned to the initial position after the surgeries, what made the symptoms of patients worsen? This issue seems more attractive to the readers.
Round 2
Reviewer 2 Report
Unfortunately, the authors did not address the suggestions sufficiently. Thus, I cannot accept the manuscript in the current version.
Reviewer 3 Report
Thank you for your reply. However, the rationale for describing the relationship between DBS surgery and SDH seems insufficient. What made the authors describe "specifically" the DBS surgery accounting for this issue? From this point of view, I, again, suggest that the main issue of this article should be the disappearance of the DBS effects after SDH. How about the incidence of SDE/SDH by other brain surgery in your own hospital? At least, how about the incidence rate of SDE/SDH by DBS and coagulation surgeries in your hospital? I still have a concern that just a two-case report on SDE/SDH by DBS surgery is important enough for readers.
